# Ecological and phylogenetic components of flatfish ectoparasites (Pleuronectiformes: Paralichthyidae) from the Southern Gulf of Mexico

Lilia C. Soler-Jiménez[1]*, Frank A. Ocaña[1,2], David I. Hernández-Mena[3], Oscar A. Centeno-Chalé[1], Ma. Leopoldina Aguirre-Macedo[1], Víctor M. Vidal-Martínez[1]

1 Aquatic Pathology Laboratory, Centro de Investigación y de Estudios Avanzados del Instituto Politécnico Nacional (CINVESTAV–IPN) Unidad Mérida, Mérida, Yucatán, México, 2 Escuela Nacional de Estudios Superiores Unidad Mérida, Universidad Nacional Autónoma de México, Tablaje Catastral N˚6998, Ucú, Yucatán, México, 3 Instituto de Biología, Universidad Nacional Autonoma de México (UNAM), Mexico City, México

* catesoler@hotmail.com

## Abstract

For many years, parasite ecologists have debated the relative importance of phylogeny and ecology as drivers of parasite community structure. Here, we address this issue using data on the metazoan ectoparasite communities of different flatfish species. Twenty species of flatfish were collected along the continental shelf of the Southern Gulf of Mexico and examined for ectoparasites. Eight flatfish species were parasitized by at least one ectoparasite species. In total, 326 ectoparasites, representing 11 species (4 monogeneans, 4 copepods, 1 isopod, 1 branchiurid, and 1 leech) were removed from 1622 hosts examined. The highest prevalence (37.5%) occurred in *Bomolochus* sp1 from *Trinectes maculatus*, while the lower (0.1%) for *Argulus* sp., *Gnathia* sp. and *Trachellobdella lubrica* from *Cyclopsetta chittendeni*. Changes in the ectoparasite community structure per host species and region were evaluated using a Permutational Multivariate Analysis of Variance and represented by a multidimensional scaling analysis. Significant differences in the parasite species composition among regions and hosts were detected, but no significant interaction between regions and hosts occurred. A multivariate pairwise *t-test* detected significant differences in the parasite infracommunities between the Yucatan Shelf and the other two regions; in addition, significant differences were detected between *C. chittendeni* and the *Syacium* species as well as between *Ancylopsetta dilecta* and *Syacium papillosum*. In this case, there is no relationship between the patterns of ectoparasitic community structure and the inherent phylogenetic affinity of the hosts; instead, the variations in ectoparasitic communities are determined by a regional ecological component.

**Data Availability Statement:** All relevant data for this study are publicly available from the Zenodo repository (https://zenodo.org/records/11123074).

**Funding:** This work was support from the National Council of Science and Technology of Mexico - Mexican Ministry of Energy - Hydrocarbon Trust, project (201441). This is a contribution of the Gulf of Mexico Research Consortium (CIGoM).

**Competing interests:** The authors have declared that no competing interests exist.

# Introduction

For many years, parasite ecologists have debated the relative importance of phylogeny and ecology as parasite community structure drivers [1–5]. In this sense, the phylogenetic component of the parasite fauna is those species that presumably have had a long co-evolutive relationship with the host. Examples of this component are those highly specific parasite species such as some monogeneans or adult cestodes of stingrays [6,7]. In contrast, the ecological component of the parasite fauna is those generalist parasites that can be acquired trophically or by active transmission, such as the cercarial stages of digeneans or larval stages of cestodes [8]. One argument is that the phylogenetic position of a host species is a better predictor of parasitic species richness than the ecological component of the parasite fauna because it covers much of its history of exposure to parasites [4]. Thus, phylogenetically related hosts are expected to harbor phylogenetically related parasites, producing a similar parasitic community structure inherited from their common ancestors through co-speciation. Many researchers have applied comparative methods to control the phylogenetic bias and determine the relative contribution of host ecology to parasite community structure [1,2,9–11]. For example, Alarco and Timi [5] compared the parasitic community structure of three sympatric flounder species. They chose closely related host species to neutralize phylogenetically inherited ecological, morphological and physiological characteristics (i.e., habitat, general shape) and a shared evolutionary history (i.e., biogeographic area of origin). Therefore, any significant change from this general homogeneous pattern could be interpreted as the consequence of ecological filters preventing homogeneous infection across host species. One of the host ecological traits identified as a critical determinant in the parasitic community structure, among many others (e.g., size and age, habitat, diet, trophic level, schooling behavior, population size, and density), is the geographical distance of the localities where the hosts are collected [10–13]. Ecologically, species can respond to Tobler's first law of geography: "Things that are closer to each other are more similar than those further away." Thus, it would be expected that in nearby locations, the similarity of parasite communities would be high, decreasing with increased geographical distance. The relative change in the contribution of the phylogenetic or ecological components of the parasite fauna to similarity with the increase in geographical distance is still a matter of debate [14–18].

During the last 20 years, the Mexican oil company PEMEX has provided financial support for environmental studies along the continental shelf of the Southern Gulf of Mexico (SGoM). Within the ichthyofauna samplings, 20 Pleuronectiform species (or flatfish) have been collected, which according to Marques et al. [19] are a convenient model for studying biogeography and host-parasite interactions since they are a monophyletic group [20,21] with different ecological strategies and life-history patterns. In Mexico, Pleuronectiforms are an abundant group of fish (with approximately 127 species in this order), some with high commercial value [22]. Despite this, biological knowledge of the parasitic fauna of these flatfish is scarce in Mexico. The few studies on Pleuronectiforms in the Gulf of Mexico have focused on endoparasitic helminths from specific flatfish species or parasite species [23–28]. In this study, we focused on ectoparasites of all flatfish species collected from the continental shelf of the SGoM during several oceanographic cruises, using the ectoparasitic community structure to compare the relative contribution of the phylogenetic and ecological components to the similarity of the parasite communities. Thus, we would expect that if the phylogenetic component of the ectoparasite communities dominates the species composition within and between localities, they also should be very similar among highly phylogenetically related host species throughout the study area. Consequently, two objectives were set for this work: 1) To provide a detailed survey of ectoparasites of flatfish collected in the SGoM, and 2) To evaluate the relative contribution

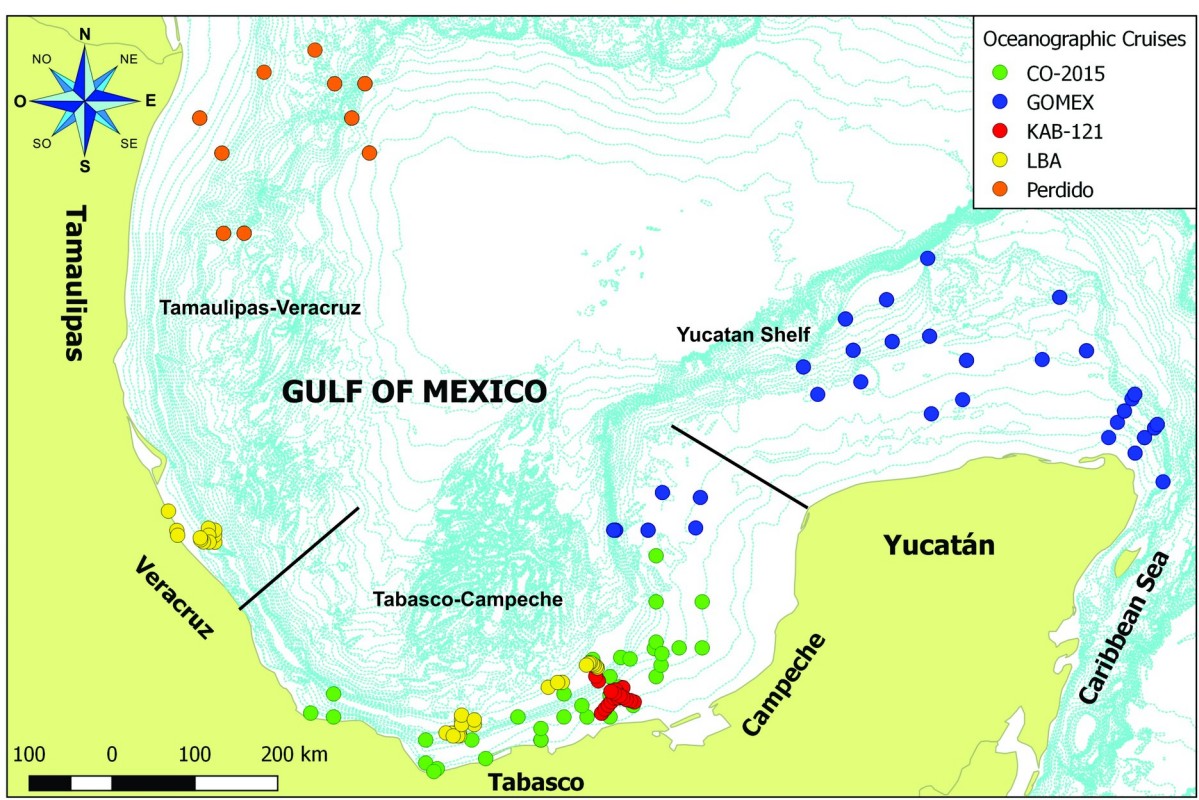

**Fig 1. Sampling stations of the 18 oceanographic cruises carried out in different regions along the continental shelf of the Southern Gulf of Mexico (SGoM) from 2008 to 2018.**

of the phylogenetic and ecological components to the similarity of the ectoparasite communities of flatfishes throughout the SGoM.

## Materials and methods

### Fieldwork

Between 2008 and 2018, 18 research cruises were carried out in different regions of the SGoM, covering 228 sampling stations (S1 Table). This study was conducted off the coast of Tamaulipas to the Yucatan Peninsula, considering three geographical regions according to the proximity of the sampling stations following Salmerón-García et al. [29] and Quintanilla-Mena et al. [30] (Fig 1): 1) Yucatan Shelf; 2) Tabasco-Campeche, from the north-western end of the Yucatan Shelf to Coatzacoalcos, Veracruz and; 3) Tamaulipas-Veracruz, from Coatzacoalcos, Veracruz to the maritime limit with the United States.

### Study area

The large-scale circulation patterns of the Gulf of Mexico are deeply influenced by the Loop Current, nourished by the waters of the Yucatan Current, which, upon leaving the Gulf, feed the Florida Current. Together, the Yucatán Loop and Florida currents are part of the intense western border current of the North Atlantic, which at times, is a fundamental part of the subtropical gyre of the Atlantic [31]. The mesoscale circulation, operating on a spatial scale of tens to hundreds of kilometers, plays a dominant role in the oceanographic dynamics of the Gulf of

Mexico, especially in its central and deep regions. The Loop Current, which originates in the Yucatán channel and becomes a pronounced meander, releases large anticyclonic eddies 200 to 300 km in diameter and 800 to 1000 m deep [32]. These eddies influence the biogeochemical and hydrographic characteristics of the water, affecting the biological productivity and the structure of marine communities. Anticyclonic eddies generate water exchanges between the shelf and the deep-water region, leaving a marked imprint on the water layers at depths of 150 to 400 m. The interaction between anticyclonic and cyclonic eddies, such as the Campeche cyclone [33], can also give rise to important flows between the shelf and the slope [34]. In the Mexican territorial sea of the Gulf of Mexico, there are very extensive continental shelves, such as the Yucatan Shelf, also known as the Campeche Bank and regions where the shelf is very narrow (such as that of the state of Veracruz) [31]. Upwellinngs in the Gulf of Mexico driven by south and southeast winds, are more frequent during spring and summer (April—August), especially on the Tabasco, Veracruz, and Tamaulipas coasts [35]. On the Yucatán Shelf, easterly winds favor upwelling throughout the year [36,37]. Significant freshwater discharges are recorded from rivers and, less obviously, through underground and underwater discharges that emerge from the seabed, especially in coastal areas, known as underwater aquifer discharges. The Mississippi-Atchafalaya system is the primary source of discharge in the Gulf, affecting the salinity and trajectory of freshwater plumes [32]. The dispersion of these plumes is influenced by seasonal winds along the coast over the continental shelf [32], with transports to the west during autumn, winter, and spring and to the northeast in summer. Interaction with the Loop Current eddies near DeSoto Canyon also affects plume distribution. The Grijalva-Usumacinta River complex that flows into the state of Tabasco contributes about 10% of the freshwater that reaches the Gulf of Mexico, especially during the summer when the peak in rainy season precipitation occurs. The Yucatán Peninsula, lacking surface rivers, provides fresh water through underwater discharges from the aquifer, affecting salinities and contributing nutrients and pollutants to the coasts of Yucatán and Campeche [32].

## Flatfish collection

Twenty flatfish species belonging to five families were collected, and a total of 1622 fish were examined for ectoparasite infestation. All flatfish collected were individually frozen (-20 ˚C) on board in plastic bags and transported to the CINVESTAV IPN Mérida Unit for parasitological examination. Ichthyologists at the Necton Laboratory (CINVESTAV-IPN-Mérida) identified all flatfish collected.

## Parasitological analysis and community descriptors

The parasitological analysis of individual flatfish included the body surface, fins, and external cavities such as the oral cavity and gills. These organs were individually examined for ectoparasites using a dissection microscope following the procedures of Vidal-Martínez et al. [38]. Gills were removed, placed in Petri dishes with filtered marine water, and carefully examined. All metazoan ectoparasites from each host were counted in situ, removed, and preserved in 4% formalin or 96% alcohol in labeled vials for subsequent morphological or molecular taxonomic identification. All the specimens found were studied in detail using an Olympus BX-50 optical microscope (Olympus, Japan). The identification of monogenean species was based on the characteristics of the sclerotized parts (haptor and male copulatory organ), according to Yamaguti [39,40]. The study of sclerotized structures of some monogeneans was performed using the proteolytic digestion method of Harris and Cable [41]. Other specimens were stained using Gomori's trichrome or Mayer's paracarmine techniques to determine the internal features, according to the method described by Kritsky and colleagues [42]. Parasitic crustaceans

(copepods, isopods, and branchiurids) were mounted and cleared with lactophenol [43] to identify species based on morphology [44–46]. The leeches were gradually relaxed in water with increasing ethanol concentrations and eventually fixed in 70%–90% ethanol [38]. Collected leeches were identified according to Tessler et al. [47] and photographed under a stereomicroscope. The prevalence, mean abundance, and mean intensity of each parasite species were determined following Bush et al. [48]. Host-specific parasites were defined as parasites found in one host flatfish species and not reported in other fish. In contrast, nonspecific parasites were defined as parasites found in two or more host flatfish species [49].

At the component community (all the parasites in a sample of a given fish species) level, we evaluated the sample completeness for each fish species using the sample coverage estimator (Cm). Component communities were described by the Hill numbers qD of order $q = 0$ (species richness). The Hill numbers were compared using 95% confidence intervals among fish species with at least 90% sample completeness. These confidence intervals were obtained by a bootstrap method based on 200 replications. Differences were considered significant if the 95% confidence intervals did not overlap [50]. These analyses were performed in the R package iNEXT [51].

## Data analysis

**Infracommunity description and multivariate analyses.** The infracommunity was defined as all metazoan ectoparasites infecting an individual fish. Ectoparasite abundance data of selected hosts (i.e., species of parasitized hosts present in all regions and with more than five individuals in each region) were transformed square roots to reduce the contribution of highly abundant species with respect to less abundant ones, and a Bray-Curtis similarity matrix was constructed. Separate one-way Permutational Multivariate Analysis of Variance (PERMANOVA) with 9999 permutations of residuals under a reduced model [52] was used to test the null hypothesis of no differences in the ectoparasite structures among regions (three levels: Yucatán Shelf; Tabasco-Campeche; and Tamaulipas-Veracruz) and among hosts (four levels: *Ancylopsetta dilecta*; *Ciclopsetta chittendeni*; *Syacium papillosum*; and *Syacium gunteri*). Likewise, the analysis was carried out separately for host-specific and nonspecific parasite species. For this analysis, the host size and the bottom depth were introduced as covariates since these two variables have been reported to affect the parasite composition of flounder species [5,53]. When PERMANOVA detected differences, t-test pairwise comparisons were used to determine which factor level differed. Resemblance patterns among ectoparasite infracommunities were represented on multidimensional metric scaling (MDS) using the distance among centroids of the host-region combination.

An analysis of similarities percentages (SIMPER) was used to detect the ectoparasite species that contributed more than 10% to the observed differences among hosts and regions. For the latter analyses, similarity percentages between ectoparasite structures were calculated from abundance values among all possible pairs of individual fish (infracommunities) within and between host species and regions and expressed as an averaged similarity for all parasite species. All statistical analyses were performed using Primer-e v7/ PERMANOVA [54,55]. Further, we tested the decay of parasite similarity over distance for all hosts. To this end, the distance matrix between sampling stations was calculated using the QGIS software and was correlated with the Bray-Curtis similarity matrix using the RELATE routine.

Considering that sample data came from different sampling years, a further analysis of similarities (ANOSIM) was performed, and no significant year effect was found for variations in the structure of the ectoparasitic community of the selected host species (S3 Table).

**Phylogenetic and distance analysis.** An analysis of the phylogenetic relationships among the parasitized host species was carried out to test how closely related the flatfish species are so that hosts phylogeny can be considered a factor explaining ectoparasite community structure. Thus, for the analysis fish from the families Achiridae, Bothidae, Cyclopsettidae, and Paralichthyidae distributed in the North Atlantic and the Gulf of Mexico were selected, all of which had published sequences of cytochrome oxidase subunit 1 (*cox 1*) in Genbank. It should be noted that all fish species included in this study were represented in the phylogenetic analysis, and the only gene that has been sequenced for all of them is Cox 1. These sequences were aligned in ClustalW [56] using the online site http://www.genome.jp/tools/clustalw/ with the parameters "SLOW/ACCURATE" and weight matrix "CLUSTALW (for DNA)." Aligned sequences were not trimmed. A substitution model was calculated in jModelTest v2 using the aligned matrix [57] to obtain a phylogenetic hypothesis using the Maximum Likelihood method (ML); this analysis was implemented in RAxML v. 7.0.4 [58], with 1,000 repetitions Bootstrap (Bt). Subsequently, to check if the ectoparasite community structure of each host species is more similar between closely related hosts than with more phylogenetically distant hosts, two analyses were carried out to group the fish based on the parasites species shared: one of similarity and another of parsimony. For these analyses, a matrix of taxon per character was built in Mesquite [59], in which the taxa were the fish species, the characters were the parasite species, and the character states were coded as 0 and 1, where 0 means absence and 1 presence of the parasite. The similarity analysis was phylogenetically analyzed by UPGMA (unweighted pair group method with arithmetic mean) implemented in MEGA X [60] with 100 bootstrap repetitions, where the distances were calculated using the number of differences method. The parsimony analysis was performed using the Tree-Bisection-Regrafting (TBR) algorithm and 100 bootstrap repetitions in TNT 1.1 [61].

## Results

### General results

A total of 1622 flatfish hosts were examined, representing 20 species. One hundred seventy-seven individual flatfish belonging to 8 species were parasitized. No ectoparasites were found on the remaining 12 flatfish species (S2 Table). A total of 326 ectoparasites belonging to 11 species were recorded, comprising 4 monogeneans, 4 copepods, 1 larval stage of isopod, 1 larval branchiurid and 1 leech (Table 1). The host species with the highest ectoparasite species number was *S. papillosum*, 8 of the 11 registered species. In contrast, *Citharichthys spilopterus* and *Cyclopsetta fimbriata* only harbored one ectoparasite species each (Table 1).

Parasites identified as host-specific were all adult-stage monogeneans (Table 1). Nonspecific parasites such as the copepods *Cresseyus* sp1. and *Caligus* sp1. were present on most of the host species parasitized, also being the most abundant. The highest prevalence (37.5%) occurred in *Bomolochus* sp1 from *Trinectes maculatus*, while the lower (0.1%) for *Argulus* sp., *Gnathia* sp. and *Trachellobdella lubrica* from *C. chittendeni*. The prevalence, mean abundance, and mean intensity values for each parasite species per host species are shown in Table 1.

The analysis of similarities (ANOSIM) for the sampling dates showed no relevant year effect on the structure of the ectoparasitic community of the different flatfish species collected on different cruises (S3 Table).

At the component community level, sampling coverage was >90% for each fish species, except for *C. fimbriata* (76%) and *C. spilopterus* (85%). According to the Hill numbers (q = 0), the diversity of parasites was significantly higher in *S. papillosum* than in the *S. gunteri*, *T. ventralis*, *T. maculatus*, *C. fimbriata* and *C. spilopterus*. The only host species that reach asymptote

**Table 1. Flatfish ectoparasite species recorded from different host and sites in the Southern Gulf of Mexico (SGoM).**

| Ectoparasite species | Hosts | N | Infested hosts | Abundance | Mean abundance ± SD | Prevalence (%) | Mean intensity ± SD |
|---|---|---|---|---|---|---|---|
| **Monogenea** | | | | | | | |
| Ancyrocephalidae gen. sp.* | *Ancylopsetta dilecta* | 42 | 2 | 2 | 0.05±0.39 | 5.4 | 1.0 |
| *Euryhaliotrema* sp.* | *Ancylopsetta dilecta* | 42 | 1 | 1 | 0.03±0.23 | 2.7 | 1.0 |
| *Microcotyle* sp.* | *Syacium papillosum* | 450 | 3 | 5 | 0.01±0.14 | 0.7 | 1.7±0.58 |
| *Paraneoheterobothrium papillosum.** | *Syacium papillosum* | 450 | 28 | 43 | 0.10±0.44 | 6.2 | 1.5±0.96 |
| **Copepoda** | | | | | | | |
| *Bomolochus* sp1. | *Trichopsetta ventralis* | 171 | 2 | 2 | 0.01±0.11 | 1.2 | 1.0 |
| | *Trinectes maculatus* | 16 | 6 | 10 | 0.63±0.96 | 37.5 | 1.7±0.82 |
| *Cresseyus* sp1. | *Ancylopsetta dilecta* | 42 | 5 | 10 | 0.27±1.78 | 13.5 | 2.0±1.41 |
| | *Citharichthys spilopterus* | 19 | 3 | 6 | 0.32±0.82 | 15.8 | 2.0±0.71 |
| | *Cyclopsetta fimbriata* | 23 | 1 | 1 | 0.04±0.21 | 4.3 | 1.0 |
| | *S.yacium gunteri* | 146 | 9 | 12 | 0.08±0.34 | 6.2 | 1.3±0.41 |
| | *Syacium papillosum* | 450 | 68 | 139 | 0.31±1.04 | 15.1 | 2.0±1.90 |
| *Caligus* sp1. | *Cyclopsetta chittendeni* | 695 | 19 | 23 | 0.03±0.21 | 2.7 | 1.2±0.41 |
| | *Syacium gunteri* | 146 | 1 | 2 | 0.01±0.16 | 0.7 | 2.0 |
| | *Syacium papillosum* | 450 | 9 | 11 | 0.02±0.18 | 2.0 | 1.2±0.44 |
| | *Trinectes maculatus* | 16 | 1 | 1 | 0.06±0.25 | 6.3 | 1.0 |
| *Acanthochondria* sp. | *Ancylopsetta dilecta* | 42 | 8 | 20 | 0.54±3.42 | 21.6 | 2.5±1.93 |
| | *Cyclopsetta chittendeni* | 695 | 9 | 11 | 0.02±0.15 | 1.3 | 1.2±0.44 |
| | *Syacium papillosum* | 450 | 1 | 1 | 0.00±0.05 | 0.2 | 1.0 |
| | *Trichopsetta ventralis* | 171 | 1 | 1 | 0.01±0.08 | 0.6 | 1.0 |
| **Branchiura** | | | | | | | |
| *Argulus* sp.[L] | *Cyclopsetta chittendeni* | 695 | 1 | 1 | 0.00±0.04 | 0.1 | 1.0 |
| | *Syacium papillosum* | 450 | 3 | 3 | 0.01±0.08 | 0.7 | 1.0 |
| **Isopoda** | | | | | | | |
| *Gnathia* sp.[L] | *Cyclopsetta chittendeni* | 695 | 1 | 1 | 0.00±0.04 | 0.1 | 1.0 |
| | *Syacium gunteri* | 146 | 1 | 1 | 0.01±0.08 | 0.7 | 1.0 |
| | *Syacium papillosum* | 450 | 5 | 8 | 0.02±0.19 | 1.1 | 1.6±0.89 |
| **Hirudinea** | | | | | | | |
| *Trachellobdella lubrica* | *Ancylopsetta dilecta* | 42 | 1 | 1 | 0.03±0.23 | 2.7 | 1.0 |
| | *Cyclopsetta chittendeni* | 695 | 1 | 1 | 0.00±0.04 | 0.1 | 1.0 |
| | *Syacium papillosum* | 450 | 6 | 6 | 0.01±0.11 | 1.3 | 1.0 |
| | *Trichopsetta ventralis* | 171 | 1 | 1 | 0.01±0.08 | 0.6 | 1.0 |

* Parasites identified as host-specific. SD = standard deviation. N = total number of individuals collected per species.

[L] Larval stage.

was *S. papillosum*. Therefore, the analyzes were based on observed ectoparasite communities, but they partially represent the ectoparasite diversity of flatfish.

## Ectoparasite variations among hosts and regions

A significant potential effect of host size and depth in parasite species composition was detected. However, this effect seemed to be constant among regions and hosts since there were no significant interactions between host size and depth or with any of the other factors (Table 2). Also, significant differences in the parasite species composition among regions and hosts were detected (Table 2). A multivariate pairwise *t-test* detected significant differences in

**Table 2. PERMANOVA results of ectoparasite composition of four flatfish hosts and three regions of the SGoM.**

| Data | Source | df | SS | MS | Pseudo-F | Pperm |
|---|---|---|---|---|---|---|
| All parasite species* | Size (S) | 1 | 6782.2 | 6782.2 | 2.5186 | **0.041** |
| | Depth (D) | 1 | 19109 | 19109 | 7.0961 | **0.001** |
| | Region (R) | 2 | 45056 | 22528 | 8.3658 | **0.001** |
| | Host (H) | 3 | 75786 | 25262 | 9.3812 | **0.001** |
| | S x D | 1 | 778.32 | 778.32 | 0.28903 | 0.897 |
| | S x R | 2 | 4681.7 | 2340.8 | 0.86928 | 0.519 |
| | S x H | 3 | 6851.4 | 2283.8 | 0.8481 | 0.573 |
| | D x R | 2 | 10242 | 5121.1 | 1.9018 | 0.057 |
| | D x H | 3 | 9401.1 | 3133.7 | 1.1637 | 0.317 |
| | R x H | 6 | 17793 | 2965.5 | 1.1013 | 0.35 |
| | Residuals | 125 | 3.37E+05 | 2692.8 | | |
| Nonspecific parasite species | Size (S) | 1 | 1176.6 | 1176.6 | 1.8395 | 0.118 |
| | Depth (D) | 1 | 5936.6 | 5936.6 | 9.282 | **0.001** |
| | Region (R) | 2 | 9462.1 | 4731.1 | 7.397 | **0.001** |
| | Host (H) | 3 | 23232 | 7744.1 | 12.108 | **0.001** |
| | S x D | 1 | 99.32 | 99.32 | 0.155 | 0.958 |
| | S x R | 2 | 931.18 | 465.59 | 0.728 | 0.605 |
| | S x H | 3 | 2391 | 797.14 | 1.246 | 0.281 |
| | D x R | 2 | 2815.3 | 1470.6 | 2.201 | 0.053 |
| | D x H | 3 | 2880.6 | 960.19 | 1.501 | 0.162 |
| | R x H | 6 | 5012.1 | 835.34 | 1.306 | 0.215 |
| | Residuals | 125 | 79951 | 639.61 | | |
| Host-specific parasite species | Size (S) | 1 | 787.04 | 784.04 | 3.021 | 0.067 |
| | Depth (D) | 1 | 197.32 | 197.32 | 0.757 | 0.41 |
| | Region (R) | 2 | 3838 | 1919 | 7.365 | **0.001** |
| | Host (H) | 3 | 950.31 | 316.77 | 1.216 | 0.273 |
| | S x D | 1 | 37.78 | 37.78 | 0.145 | 0.847 |
| | S x R | 2 | 381.41 | 190.7 | 0.732 | 0.524 |
| | S x H | 3 | 76.59 | 25.53 | 0.098 | 0.986 |
| | D x R | 2 | 450.03 | 225.01 | 0.863 | 0.45 |
| | D x H | 3 | 163.19 | 54.39 | 0.209 | 0.953 |
| | R x H | 6 | 728.92 | 121.49 | 0.466 | 0.883 |
| | Residuals | 125 | 32568 | | | |

the composition of the parasite communities between the Yucatán Shelf and the other two regions (to all ectoparasite species and host-specific/nonspecific ectoparasite species) (Table 3). In addition, significant differences were detected between *C. chittendeni* and the *Syacium* species, as well as between *A. dilecta* and *S. papillosum*, with all species (Fig 2) (Table 3). For nonspecific parasite species, depth is an important explanatory variable. Furthermore, significant differences between hosts were detected for all combinations except *S. gunteri* and *S. papillosum* (Table 3). The SIMPER analysis showed that the most remarkable dissimilarity in the ectoparasite community structures occurred between *S. gunteri* and *C. chittendeni* (Av. Diss = 95.7). The copepod *Cresseyus* sp1. was the species that contributed with the highest percentages to the dissimilarity between almost all paired hosts (Table 4) and regions (Table 5).

**Table 3. P-values of multivariate pairwise t-test comparison between regions (A) and hosts (B).**

| A) Regions | All species | | | Nonspecific parasite species | | | Host-specific parasite species | | |
|---|---|---|---|---|---|---|---|---|---|
| | Tamaulipas-Veracruz | Tabasco-Campeche | | Tamaulipas-Veracruz | Tabasco-Campeche | | Tamaulipas-Veracruz | Tabasco-Campeche | |
| Tabasco-Campeche | 0.248 | | | 0.216 | | | 0.775 | | |
| Yucatan Shelf | **0.003** | **0.001** | | **0.004** | **0.001** | | **0.032** | **0.001** | |
| B) Hosts | | | | | | | | | |
| | *A. dilecta* | *C. chittendeni* | *S. gunteri* | *A. dilecta* | *C. chittendeni* | *S. gunteri* | | | |
| *C. chittendeni* | 0.053 | | | **0.023** | | | | | |
| *S. gunteri* | 0.067 | **0.001** | | **0.029** | **0.001** | | | | |
| *S. papillosum* | **0.001** | **0.001** | 0.574 | **0.001** | **0.001** | 0.592 | | | |

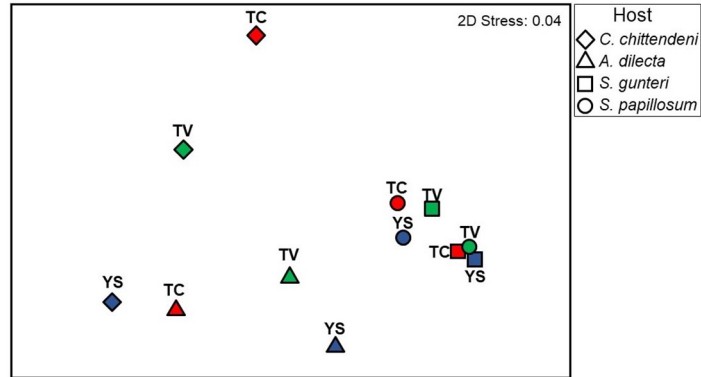

**Fig 2. MDS ordination of the ectoparasite composition based on the distance among centroids for the combinations of Regions and Hosts.** Each region is identified with an acronym and color pattern: Tamaulipas-Veracruz (T-V; green); Tabasco-Campeche (T-C; red); Yucatan Shelf (YS; blue).

**Table 4. SIMPER results showing the average dissimilarities (Av. Diss.) and the taxa that contributed (Cont%) with more than 10% to the observed differences between any pair of compared hosts.**

| Host | *S. gunteri* | | *S. papillosum* | | *C. chittendeni* | |
|---|---|---|---|---|---|---|
| *S. papillosum* | Av. Diss. = 48.6 | | | | | |
| | **Taxa** | **Cont%** | | | | |
| | *Cresseyus* sp1. | 52.7 | | | | |
| | *P. papillosum* | 20.4 | | | | |
| | *Caligus* sp. | 10.7 | | | | |
| *C. chittendeni* | Av. Diss. = 95.7 | | Av. Diss. = 88.1 | | | |
| | **Taxa** | **Cont%** | **Taxa** | **Cont%** | | |
| | *Cresseyus* sp1. | 45.8 | *Cresseyus* sp1. | 41.8 | | |
| | *Caligus* sp. | 31.6 | *P. papillosum* | 26.2 | | |
| | *Acanthocondria* sp. | 14.1 | *Acanthocondria* sp. | 15.9 | | |
| *A. dilecta* | Av. Diss. = 78.6 | | Av. Diss. = 82.3 | | Av. Diss. = 90.8 | |
| | **Taxa** | **Cont%** | **Taxa** | **Cont%** | **Taxa** | **Cont%** |
| | *Cresseyus* sp1. | 42.6 | *Cresseyus* sp1. | 32.3 | *Acanthocondria* sp. | 44.9 |
| | *Acanthocondria* sp. | 42.1 | *P. papillosum* | 23.1 | *Caligus* sp. | 30.9 |
| | | | *Acanthocondria* sp. | 14.5 | *Cresseyus* sp1. | 17.1 |
| | | | Ancyrocephalidae | 12.4 | | |

**Table 5. SIMPER results showing the average dissimilarities (Av. Diss.) and the taxa that contributed (Cont%) with more than 10% to the observed differences between any pair of compared regions.**

| Region | Yucatan Shelf | | Tabasco-Campeche | |
|---|---|---|---|---|
| **Tabasco-Campeche** | Av. Diss. = 67.5 | | | |
| | **Taxa** | **Cont%** | | |
| | *Cresseyus* sp1. | 42.6 | | |
| | *P. papillosum* | 20.7 | | |
| | *Caligus* sp. | 17.3 | | |
| **Tamaulipas-Veracruz** | Av. Diss. = 51.0 | | Av. Diss. = 60.0 | |
| | **Taxa** | **Cont%** | **Taxa** | **Cont%** |
| | *Cresseyus* sp1. | 52.6 | *Caligus* sp. | 39.8 |
| | *P. papillosum* | 24.2 | *Acanthocondria* sp. | 31.4 |
| | | | *Cresseyus* sp1. | 13.5 |

For the four fllatfish hosts, no decay pattern of ectoparasite similarity over distance was found (*S. gunteri*: Rho = -0.06, P = 0.733; *S. papillosum*: Rho = 0.07, P = 0.203; *C. chittendeni*: Rho = 0.02, P = 0.387; *A. dilecta*: Rho = 0.29, P = 0.142).

## Phylogenetic relationships of pleuronectiforms and their ectoparasitic communities

The 8 parasitized pleuronectiform species in this analysis belong to four families: Achiridae, Bothidae, Cyclopsettidae, and Paralichthyidae. The aligned matrix with sequences from these 8 species contained additional sequences from another 32 species of pleuronectifom from the same families. The matrix had a length of 677 bp. The maximum likelihood tree had a value of -9769.82 (Fig 3A). As a result of our analysis, Achiridae and Paralichthyidae formed monophyletic groups. At the same time, the Bothidae species nested within a clade that also grouped all the Cyclopsettidae species. Therefore, none of these two families appears to be a monophyletic group. In particular, the parasitized species in this study formed monophyletic groups with their respective congeners. In Achiridae, *Trinectes maculatus* was grouped with *Trinectes paulistanus* and *Trinectes inscriptus* (Bt = 77). In Bothidae, *Trichopsetta ventralis* was the sister species of a species of the genus *Bothus* (Bt = 75). In Paralichthyidae, *A. dilecta* was grouped with *Ancylopsetta ommata* (Bt = 95); *Citharichthys spilopterus* was grouped with *Citharichthys gilberti* (Bt = 95) in a monophyletic clade that contained the rest of the species of genus *Citharichthys* (Bt = 58); *C. chittendeni* and *C. fimbriata* were grouped with *Cyclopsetta querna* and *Cyclopsetta panamensis* (Bt = 98); *S. gunteri* and *S. papillosum* were sister species to each other (Bt = 99), which in turn formed a well-supported clade with *Syacium micrurum* and *Syacium maculiferum* (Bt = 91). In the UPGMA analysis (Fig 3B), the ectoparasite community structures of *Citharichthys spilopterus* and *C. fimbriata* were identical (Bt = 100) and in turn they were more similar to *S. gunteri* (Bt = 60). In contrast, *T. maculatus* was more similar to *T. ventralis* (Bt = 56). *C. chittendeni* was more similar to *S. papillosum* (Bt = 63), while *A. dilecta* was the most different fish in terms of its ectoparasite community structure. Five equally parsimonious trees were obtained for the parsimony analysis (Fig 3C) (length = 16, consistency index = 0.583333, and the retention index = 0.642857 for parsimony-informative sites). *Syacium papillosum* was grouped with *C. chittendeni* (Bt = 76), and in turn grouped with *A. dilecta* (Bt = 42). On the other hand, *Citharichthys spilopterus* was grouped with *C. fimbriata* (Bt = 32), and these were grouped with *S. gunteri* (Bt = 25). Finally, *T. maculatus* was grouped with *T. ventralis* (Bt = 43). It is worth mentioning that neither analysis showed any topological

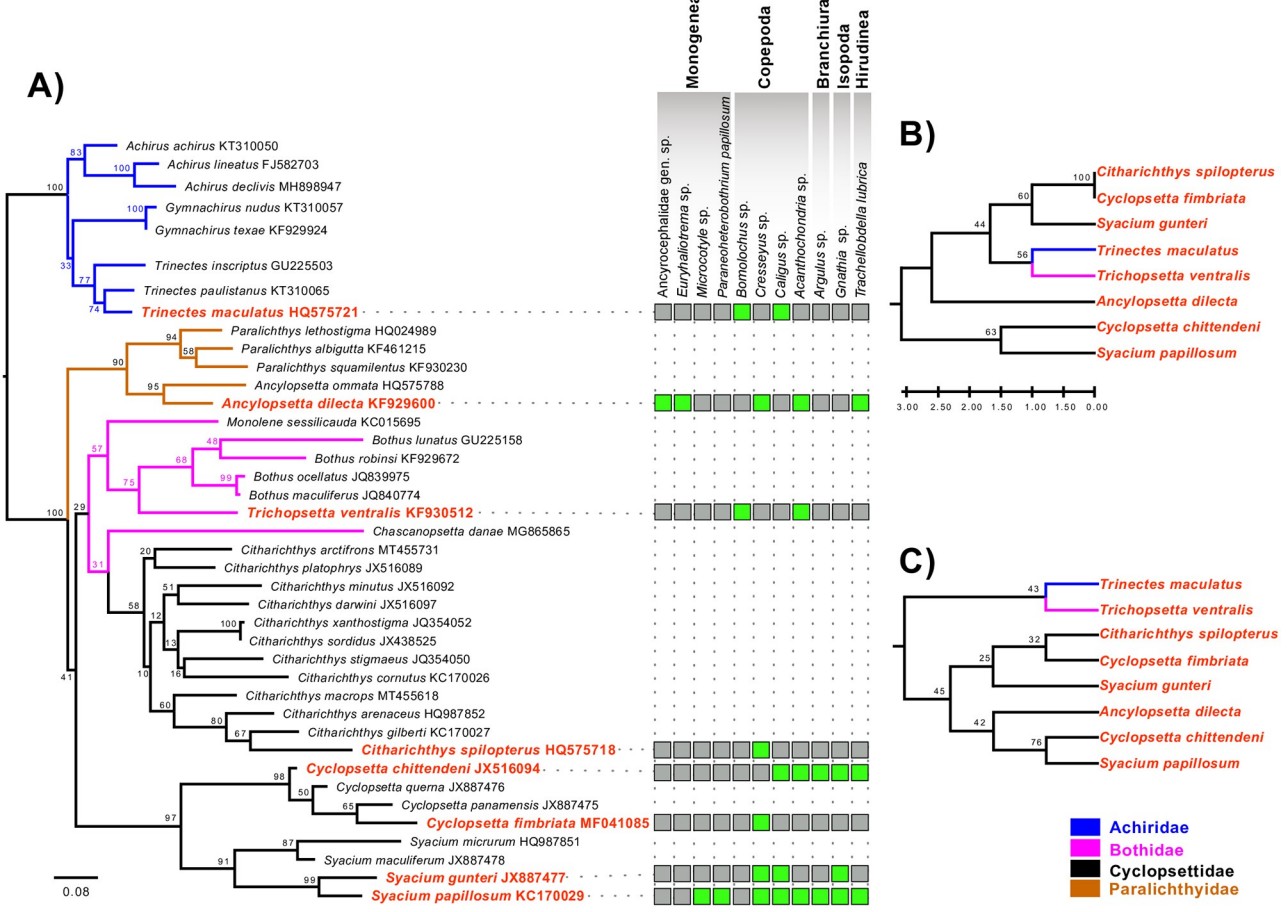

**Fig 3. Phylogenetic relationships of pleuronectiforms and their ectoparasitic communities. A)** The maximum likelihood tree; **B)** Similarity analysis UPGMA (unweighted pair group method with arithmetic mean); and **C)** Parsimony analysis.

congruence with the phylogenetic relationships of the fish and that, in general, the bootstrap support values were low.

## Discussion

In the present study, the patterns in the ectoparasitic community of 20 flatfish species throughout the SGoM were analyzed, with the expectation that these would be similar in species of phylogenetically closely related hosts. This expectation is based on the idea that phylogenetic convergence between closely related host species with similar ecological characteristics and living in sympatry could harbor similar parasite faunas, decreasing in similarity with increased phylogenetic distance among hosts [62–64]. However, the phylogenetic analysis found no evidence of congruent clustering associated with the hosts' inherent phylogenetic affinity when looking for a relationship between hosts and their ectoparasite communities. If the ectoparasitic communities were the result of host phylogenetic diversification, it would be expected that species of the same genus would be grouped (*S. gunteri* with *S. papillosum* or *C. chittendeni* with *C. fimbriata*), but this was not the case (Fig 3). Thus, our interpretation is that the flatfishes, independently of the genus, acquired available ectoparasites in the environment, i.e., a basic ecological process mediated by the host's depth, size, and geographical structure.

Similar results were obtained by Osuna-Cabanillas et al. [65], who claim that phylogenetic relatedness and sympatry are weak determinants of the structure and composition of parasite communities in carangid species. Likewise, Poulin [64] observed that, within a single fish family, the decay in similarity of parasite communities between fish species is not explained by phylogenetic relationships.

On the other hand, similarity analyzes indicate that there were no significant differences in the structure of ectoparasitic communities between *Syacium* species, which could suggest that the phylogenetic affinity of the hosts is involved. However, it must be considered that nonspecific parasite species are responsible for the similarities between the ectoparasitic structure in these two host species. Osuma-Cabanilla et al. [65] explain that the observed differences in their study were influenced by non-common parasites (host-specific) because when the analysis was restricted to common parasites species, some fish showed homogeneous parasite infracommunities. This homogeneity may happen because common, nonspecific parasites are not inherited from a common ancestor and are potentially available for all host species [66]. By contrast, host-specific ectoparasites species, such as *Paraneoheterobothrium papillosum*, contribute significantly to the percentage of dissimilarity of the ectoparasite communities. These host-specific species typify the ectoparasite community of its host. Alarco and Timi [5] also compared the component communities of three related pleuronectiform species, observing in the same way that all host-specific parasites were of little importance as contributors to abundance and species richness in three flounder species or to the similarity between host species. There is evidence that host specificity is a central aspect of symbiont life history with broad ecological and evolutionary implications [65,67]. However, host-specificity had a minor role in this case due to the low number of host-specific parasites.

It is a fact that a certain degree of similarity in their parasite community structures should be expected for related host species [2,10,11,19,49], given that related host species can harbor related parasites inherited from their common ancestors through co-speciation [68,69]. However, some aspects of parasite biology are more labile and not subject to phylogenetic signals. For instance, the number and specific identities of host species exploited by a parasite at any given life cycle stage vary among closely related hosts. Therefore, the local availability of host species can influence estimates of how many hosts are used or which host species are preferred [4,70]. Krasnov et al. [71] explained that the phylogenetic signal in the number of host species used by a given ectoparasite species, as well as the average abundance achieved by an ectoparasite species on their hosts, is much stronger when based on continental-scale data than on regional-scale data. That is, when examined at the appropriate scale, the ecological traits of parasites generally show clear phylogenetic signals. Thus, if the analysis is carried out on a larger scale, the differences in the ectoparasite community structures between host species could be compensated and not mask the actual phylogenetic signal.

Regarding the conformation of the ectoparasite communities throughout the study area, the similarities in the ectoparasite community structure in the Tamaulipas-Veracruz and Tabasco-Campeche regions, and at the same time, their differentiation concerning the Yucatan Shelf, can be attributed to the composition of parasitized hosts in each region. While Tamaulipas-Veracruz and Tabasco-Campeche presented a very similar host species composition (both in relative abundance and in species richness), the set of host species of the Yucatan Shelf is composed mainly of individuals of the species *S. papillosum*. Marques et al. [19] proposed that in areas with similar host composition, the ectoparasite structure is also expected to be similar. In addition, the distribution, abundance, and host behavior, alone or all together, also contribute to their species composition differentiation. On the other hand, several mechanisms can act, alone or in combination, to reduce the similarity in species composition among communities with increasing geographical distance [72,73]. A decrease in the similarity of

their environmental conditions, geographical barriers, or both can lead to species sorting in geographical space [74]. Therefore, the ectoparasite community structure would be expected to differ in distant regions with different dynamics and characteristics, such as Tabasco-Campeche, Tamaulipas-Veracruz, and the Yucatan Shelf. However, Tamaulipas-Veracruz and Tabasco-Campeche did not fit this model. Recent studies on the Regional Ocean Modeling System analyzing cross-shelf transport associated with the presence of mesoscale anticyclones as they interact with the western shelf of the Gulf of Mexico have shown that seasonal convergence occurs as a result of an "eddy complex," an anticyclone in the western Gulf of Mexico and a cyclone in the Bay of Campeche [34], which seems to favor the connectivity between these two regions. It likely contributes to the similarity found in the ectoparasite communities.

On the other hand, the water masses in the Yucatan Shelf are more influenced by the Yucatan current that enters the Gulf from east to west and north through the Yucatan channel; when entering the Loop Current, it gives off cyclonic and anticyclonic eddies that travel towards the interior of the Gulf [75,76], which makes the circulation of the water masses in the Yucatan Shelf different from the rest of the Gulf. Additionally, the northward portion of the Loop current flow at the edge of the Yucatan continental shelf could contribute to forming a "physical barrier" that impedes the flow of ectoparasite larvae from the Yucatan Shelf to Campeche Sound. According to the above, the variability in the ectoparasite community between regions could be explained mainly by the host composition of each region, which can be determined by the landscape topography and physical conditions, which, in turn, can limit the dispersal of ectoparasite species.

## Conclusions

In conclusion, the dissimilarity of the structure of the ectoparasitic communities of flatfish in the SGoM seems to be influenced by the presence of host-specific ectoparasites, which in turn depends on the distribution of the hosts, which may be determined by the topography of the landscape and physical conditions of each region. In contrast, nonspecific parasites present broader distribution ranges according to the number of host species that they parasitize. Although, there is no evidence in this study that the phylogenetic affinity between flatfish host species is an essential component in the structure of the ectoparasitic community, it cannot be ruled out. This phylogenetic affinity may be masked by differences in the local availability of host species that do not allow us to fully estimate how many host species are used or which hosts are preferred. A broader geographic scale analysis may compensate for variation in host availability/abundance by region. Consequently, an interspecific balance variation among ectoparasitic community structure may reveal an actual phylogenetic signal.

## Supporting information

**S1 Table. Research cruises carried out in different regions along the continental shelf of the Southern Gulf of Mexico (SGoM).**
(DOCX)

**S2 Table. Flatfish hosts species collected in the Southern Gulf of Mexico (SGoM) from the 18 oceanographic cruises (2010–2018) mentioned in S1 Table.** N = total number of individuals collected.
(DOCX)

**S3 Table. ANOSIM analysis results for the sampling data.**
(DOCX)

## Acknowledgments

The authors thank Clara Vivas-Rodríguez, Gregory Arjona-Torres, Francisco Puc-Itza, and Mirella Hernández de Santillana for support with the field and laboratory work sopport. We acknowledge PEMEX for the specific request to the Hydrocarbon Found to address the environmental effects of oil spills in the Gulf of Mexico.

## Author Contributions

**Conceptualization:** Lilia C. Soler-Jiménez, Víctor M. Vidal-Martínez.

**Data curation:** Lilia C. Soler-Jiménez, Oscar A. Centeno-Chalé.

**Formal analysis:** Frank A. Ocaña.

**Funding acquisition:** Ma. Leopoldina Aguirre-Macedo.

**Investigation:** Lilia C. Soler-Jiménez, Oscar A. Centeno-Chalé.

**Methodology:** Frank A. Ocaña, David I. Hernández-Mena.

**Supervision:** Víctor M. Vidal-Martínez.

**Validation:** Frank A. Ocaña.

**Visualization:** Lilia C. Soler-Jiménez.

**Writing – original draft:** Lilia C. Soler-Jiménez.

**Writing – review & editing:** Frank A. Ocaña, David I. Hernández-Mena, Ma. Leopoldina Aguirre-Macedo, Víctor M. Vidal-Martínez.

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
