## [Decision Letter · Decision Letter 0]

31 Oct 2023

PONE-D-23-31417Ecological and phylogenetic components of flatfish ectoparasites (Pleuronectiformes: Paralichthyidae) from the Southern Gulf of Mexico

Dear Dr. Soler-Jimenez,

Thank you for submitting your manuscript to PLOS ONE. After careful consideration, we feel that it has merit but does not fully meet PLOS ONE’s publication criteria as it currently stands. Therefore, we invite you to submit a revised version of the manuscript that addresses the points raised during the review process.

Kindly address all the comments and suggestions made to the reviewers to ensure your revised version meets the standard of the journal. Please submit your revised manuscript by Dec 15 2023 11:59PM. If you will need more time than this to complete your revisions, please reply to this message or contact the journal office at plosone@plos.org. Please include the following items when submitting your revised manuscript:A rebuttal letter that responds to each point raised by the academic editor and reviewer(s). You should upload this letter as a separate file labeled 'Response to Reviewers'.A marked-up copy of your manuscript that highlights changes made to the original version. You should upload this as a separate file labeled 'Revised Manuscript with Track Changes'.An unmarked version of your revised paper without tracked changes. You should upload this as a separate file labeled 'Manuscript'.

We look forward to receiving your revised manuscript.

Kind regards,

Joshua Kamani, PhD

Academic Editor

PLOS ONE

Journal Requirements:

"This work was support from the National Council of Science and Technology of Mexico - Mexican Ministry of Energy - Hydrocarbon Trust, project (201441). This is a contribution of the Gulf of Mexico Research Consortium (CIGoM)." 

5. We note that Figures 1 and S1 in your submission contain [map/satellite] images which may be copyrighted. All PLOS content is published under the Creative Commons Attribution License (CC BY 4.0), which means that the manuscript, images, and Supporting Information files will be freely available online, and any third party is permitted to access, download, copy, distribute, and use these materials in any way, even commercially, with proper attribution. For these reasons, we cannot publish previously copyrighted maps or satellite images created using proprietary data, such as Google software (Google Maps, Street View, and Earth). For more information, see our copyright guidelines: http://journals.plos.org/plosone/s/licenses-and-copyright.

a. You may seek permission from the original copyright holder of Figures 1 and S1  to publish the content specifically under the CC BY 4.0 license.  

Additional Editor Comments :

Authors are invited to take note of the following in addition to the other comments

Lines 6-9- (Three regions……… the United States) not necessary in the abstract. Pleases delete.

Lines 12 & 13 write ‘’A. dilecta’’ and ‘’C. chittendeni’’ in full and subsequently use abbreviations for genus names. This should be reflected throughout the manuscript.

Table 1 column 7 provide the English version of the caption

Line 79 Change – …’parasitologically examined’ to ‘’examined for ectoparasite infestation’’

Line 82- Mention the method used for the identification, taxonomic keys or ??? and provide reference(s) for the method.

Lines 92-93. I am not sure if this is ideal for metazoan ectoparasites- All specimens found were studied at 1,000 x magnification using immersion oil with an Olympus BX-50 light microscope (Olympus, Japan). Please modify

Line 142 italicize (cox 1)

Line 166- use infested instead of infected

Lines 164 -168- if total flatfish species were examined, and 10 were infested, then the negative shouldn’t be 13 species. Cross check please.

Table 2- Add a row at the end of the table for the total of number of flatfish examined and the number infested

Table 3 caption- check spelling of ‘’different’’. Move ‘SD = standard deviation. N = total number of 193 individuals collected’ as footnotes under the table

Line 268- Correct -C) Parsimony ‘analysi’

Reviewers' comments:

Reviewer's Responses to Questions

**Comments to the Author**

1. Is the manuscript technically sound, and do the data support the conclusions?

Reviewer #1: Yes

Reviewer #2: Partly

Reviewer #3: No

2. Has the statistical analysis been performed appropriately and rigorously? 

Reviewer #1: No

Reviewer #2: Yes

Reviewer #3: No

3. Have the authors made all data underlying the findings in their manuscript fully available?

Reviewer #1: Yes

Reviewer #2: Yes

Reviewer #3: No

4. Is the manuscript presented in an intelligible fashion and written in standard English?

Reviewer #1: Yes

Reviewer #2: Yes

Reviewer #3: Yes

5. Review Comments to the Author

Reviewer #1: The work is interesting, as it proposes to investigate a problem that is still widely discussed in the area of parasitological ecology. However, I made some observations that should be considered by authors so that the work is suitable for publication:

1 - the authors hypothesize that the structures of ectoparasite communities are the result of the geographic variability of the SGoM continental coast or whether the patterns depend on the phylogeny of the hosts (lines 57 to 62). However, no verification of the geographic hypothesis was carried out.

2 - it is necessary for the authors to describe in more detail the region where the work was carried out, such as informing if there are seasonal changes in environmental characteristics (variables) such as water salinity, temperature, pH, productivity, ... , if there is a record of changes in the ichthyofauna over the years...

3 - explain the reason for transforming the data, instead of using raw data. Explain why Euclidean distance is used as a dissimilarity/similarity metric, instead of, for example, the Bray-Curtis index.

4 - The authors present, as one of the results, a phylogeny where the families Bothidae and Paralichthydae both appear as non-monophyletic. I imagine that this phylogeny must result from the use of a single molecular marker, as well as the length of the sequence used. Recent studies demonstrate the monophyly of these families (see Campbell et al., 2019; Atta et al., 2022). Since the analyzes depend on a well-resolved phylogeny of the hosts, I suggest that the analyzes be redone with the organization of the phylogeny of the species considered in the study being appropriate to the best resolved phylogenies.

Minor reviews:

Line 23 - change to "the ecology and phylogeny of hosts as directors";

Line 25 - change to "host species is a better predictor";

Lines 62 to 64 - this prediction is valid if the phylogeny of the hosts is the main or only factor that directs the patterns of ectoparasite community structures. However, another hypothesis was also suggested, so the prediction of this hypothesis must also be presented;

Lines 64 to 66 - the objectives are to test the hypotheses that have been suggested;

Lines 69, 70 - inform the duration of each "Research cruises";

Table 1 - the name of the last column is in Spanish, translate into English; tables do not contain the internal drawn lines;

Lines 87, 88 - Insert Vidal-Martínez and colleagues before reference [26];

Line 96 - replace "&" with "and";

Line 98 - replace "et al." by "colleagues" before reference [28];

Line 103 - Insert Bush et al. before reference [31];

Line 117, 118 - the term parasite infracommunity was defined in lines 109 and 110, therefore, the authors should remove or use this definition in lines 109, 110;

Line 142 - inform the genbank access codes (as supplementary material), as well as the number of bases used in the sequences;

Line 164 - this information has already been presented on line 79;

Table 2 - tables do not contain the internal drawn lines;

Table 3 - tables do not contain the internal drawn lines;

Line 324 - replace "shelf" with "Shelf" ;

Line 386 - correct the name of the periodical;

Figures 1 to 4 are not of good quality.

Reviewer #2: In the manuscript entitled “Ecological and phylogenetic components of flatfish ectoparasites (Pleuronectiformes: Paralichthyidae) from the Southern Gulf of Mexico”, Soler-Jiménez et al. explore ectoparasite diversity patterns using an impressive number of flatfish recovered from many vessels that sampled over an extensive region in the Southern Gulf of Mexico. The authors attempt to answer whether phylogenetic congruence or ecological differences better explain flatfish ectoparasite diversity. The abstract is a good description of methods and results, but the broader context and its implications are missing. Because of that, the study questions and aims are not clear until the reader goes over the whole introduction. Overall, I find the study interesting and the questions relevant, however, there are many points for improvement. Firstly, there are confounding factors (host geographical structure, different sample sizes, generalist vs. specialist parasites, different sampling times and seasons) that need to be accounted for. There may be various ways to do so (e.g., rarefying), and I also suggest a few other analyses that could be undertaken and potentially further support the conclusions of the study. Among these analyses are investigations of distance-decay patterns, a separate phylogenetic congruence test for specialist and generalist parasites, and the use of a different genetic marker for fish phylogenies that is more appropriate for family-level comparisons (I noticed that the flatfish phylogenetic tree has many nodes of low bootstrap support, which could make phylogenetic congruence inferences even more challenging). In addition, a correction for multiple tests (at least) in pairwise comparisons must be made (or details on how it was done must be added). I also believe a more thorough description is warranted of how the different sample sizes affect analyses and the discovery of ectoparasites recovered from each flatfish species (ultimately affecting diversity measures). In terms of text structure, the results are repetitive with the information on tables and figures. Also, while the study system and the findings are the focus, including the broader context in the discussion would be beneficial. I left detailed comments in the file attached, split by major concern points and minor points. I believe that addressing such points will help towards having stronger evidence to explain the mechanisms underpinning the flatfish ectoparasite diversity patterns observed.

Reviewer #3: Review – PONE-D-23-31417

The authors sampled a large number of pleuronectiform fish from 23 different species over a period of 12 years from different regions (grouped into three areas) for ectoparasites. The main aim was to examine the variability on ectoparasite communities in pleuronectiform fish from the area. The authors conclude that variation in these parasite communities are determined primarily by ecology rather than phylogenetics.

The study is well written and does include a large number of samples. However, I do have some concerns that I list below.

For 12 of the 23 species of pleuronectiform fishes sampled, the total sample size is less than 1 per year sampled. As such, given the high variability on prevalence and abundance of parasites in both spatial and temporal spaces, the implications of this unbalanced design need to be discussed.

Additionally, the “perceived” absence of certain parasites from some of the host species collected in “low” numbers could affect the phylogenetic signal and inflate the likelihood of making a type II error, i.e. there might be a phylogenetic signal, but it is not detected due to missing data. Making a type II error is more significant than making a type I error. This needs to be addressed, or at the very least explicitly discussed.

I suggest the authors attempt to determine whether their parasite sample for individual host species is representative of the expected ectoparasite fauna. This could be achieved by following the method outlined in Bizzarro JJ, Smith WD, Márquez-FarÌas JF, Tyminski J, Hueter RE (2009) Temporal variation in the artisanal elasmobranch fishery of Sonora, Mexico. Fish Res 97:103–117. If the species accumulation curves do not reach asymptote, this needs to be discussed, i.e. the analyses are based on observed ectoparasite communities, but might not be representative of the “true” ectoparasite diversity of hosts a, b, c, etc.

There is no description of the size range sampled for the different pleuronectiform fishes. For instance, due to sampling method (outlined in table 1), is it not possible that authors have a size bias in their sample. Given what is known of the ecology of certain flatfish species, we know that they change substrate and habitat during ontogeny (therefore also salinity and temperature). The length frequencies for each species need to be presented and assessed for signs of size bias. How might this affect the outcome of analyses should then be discussed.

It is not clear whether host identified only to genus level could be conspecific with others sampled in this study. This needs to be outlined clearly.

For the phylogenetic relationships within the order, authors added 32 species of the same families. However, there is no table listing these species and their taxonomic affinities. This is shown via a figure, but without GenBank accession numbers and taxonomic affinities provided, it is difficult to assess the adequacy of included taxa. Additionally, there is no rationale provided as to why these 32 taxa were selected. Given the apparent bias in taxa selected for phylogenetic analyses, it is possible that the trees are not similar with the “true” topology due to gaps in taxonomic sampling for phylogenetic analyses. Was the topology compared to that of a tree for the Pleuronectiformes (or similar)?

There seems to be an error in reporting prevalence in the Abstract and Results with prevalence ranging from 0.01 to 37.5%. Since fewer than 2000 hosts were sampled, I would expect the prevalence to range from 0.1 to 37.5%. Check other calculations as well.

Furthermore, there is likely a “year” effect that should be assessed. Since the study spans 12 years and we know that parasite communities fluctuate in space and time, it is not unreasonable to suggest that year of sampling could affect the results.

I don’t think that the conclusions reached match the data. From the data, as analysed, it would seem that there is no phylogenetic signal. However, there was no assessment of environmental variables in this study, therefore, authors cannot conclude that the variability on ectoparasite communities is due to ecology. I have provided some comments on other uncertainties that could cloud the outcome. Without a proper analysis of effects on specific environmental variables on ectoparasite communities, authors cannot conclude that the variation is due to the environment.

6. PLOS authors have the option to publish the peer review history of their article (what does this mean?). If published, this will include your full peer review and any attached files.

Reviewer #1: No

Reviewer #2: No

Reviewer #3: No

---

## [Author Response · Author response to Decision Letter 0]

6 May 2024

We thank the reviewers for their recommendations and comments. All questions and comments made by the reviewers and the editor were considered in the "Responses to Reviewers" document.

---

## [Decision Letter · Decision Letter 1]

9 Jun 2024

PONE-D-23-31417R1Ecological and phylogenetic components of flatfish ectoparasites (Pleuronectiformes: Paralichthyidae) from the Southern Gulf of MexicoPLOS ONE

Dear Dr. Soler-Jimenez,

Thank you for submitting your manuscript to PLOS ONE. After careful consideration, we feel that it has merit but does not fully meet PLOS ONE’s publication criteria as it currently stands. Therefore, we invite you to submit a revised version of the manuscript that addresses the points raised during the review process.

We look forward to receiving your revised manuscript.

Kind regards,

Joshua Kamani, PhD

Academic Editor

PLOS ONE

Journal Requirements:

Additional Editor Comments:

Dear Lilia

Thank you for submitting your paper to PLOS One

After a careful evaluation, I am to request you to address some comments by the reviewer before we can decide to accept your manuscript for publication

Reviewers' comments:

Reviewer's Responses to Questions

**Comments to the Author**

1. If the authors have adequately addressed your comments raised in a previous round of review and you feel that this manuscript is now acceptable for publication, you may indicate that here to bypass the “Comments to the Author” section, enter your conflict of interest statement in the “Confidential to Editor” section, and submit your "Accept" recommendation.

Reviewer #1: All comments have been addressed

Reviewer #2: (No Response)

2. Is the manuscript technically sound, and do the data support the conclusions?

Reviewer #1: Yes

Reviewer #2: Yes

3. Has the statistical analysis been performed appropriately and rigorously? 

Reviewer #1: Yes

Reviewer #2: Yes

4. Have the authors made all data underlying the findings in their manuscript fully available?

Reviewer #1: Yes

Reviewer #2: Yes

5. Is the manuscript presented in an intelligible fashion and written in standard English?

Reviewer #1: Yes

Reviewer #2: No

6. Review Comments to the Author

Reviewer #1: The authors have resolved many of the issues raised in the first revision, resulting in a much more attractive manuscript. However, some issues still need to be resolved.

In lines 66 to 70 and lines 75 to 77, the authors aim to assess the contribution of ecological components to the similarity of flatfish ectoparasite communities across the SGoM. It can be seen in the data analysis section that the depths of the regions investigated (line 176), as well as the geographical distances between them (lines 189 and 190) were considered in the analyses. Do these variables correspond to the only ecological components considered? If so, this should be clearly defined in the objectives.

Why was the similarity of host species between regions not considered as an explanatory variable?

Minor reviews:

line 167: typo error "e.i." for "i.e.";

lines 167 to 169: insert in the text the reason for transforming the raw data;

line 203: typo error "Gulf of Mexico. , and which"

Reviewer #2: The authors of the manuscript entitled “Ecological and phylogenetic components of flatfish ectoparasites (Pleuronectiformes: Paralichthyidae) from the Southern Gulf of Mexico” dedicated great effort to this revised version of the manuscript, and to my view the new version is a significant improvement from the original submission. Relevant analyses were added and the introduction and discussion now clearly identify the broader context in which this work is placed. My main concerns are now (1) a number of grammatical errors and some poor sentences that should be corrected and improved before publication; and (2) that there still seems to be some emphasis in understanding co-evolution between hosts and parasites when the authors said in the response letter that their data lacks the resolution for a formal test of the relationship between host and parasite phylogenies. I made a few suggestions below to ‘tone down’ the co-phylogeny emphasis, as well as some suggestions for grammar/sentence structure (although I did not make an exhaustive search of all the errors, so I recommend the authors proof-read their manuscript prior to publication). I should also mention I appreciate that the authors now analysed ecological components using depth and host size. Their results were significant, so I think at least a sentence about it should be in the discussion. In L 351, does “basic ecological processes” mean depth, size, and host geographical structure? If so, these three factors could be simply named there. Finally, regarding PERMANOVA and sample sizes, the reason why I mentioned sample sizes and rarefaction is not because of the statistical analysis used, but because of the probability of finding parasites. If a parasite has a 1% prevalence and you collect 5 fish of species A but 100 fish of species B, you will most likely find it only on species B. However, the authors did include a rarefaction analysis in their new version and clearly explained that the diversity in most cases did not level off, which in my opinion is all that is needed.

Overall, I think the authors carefully considered all the comments made by reviewers and addressed all of the suggestions either by modifying the manuscript or by explaining their rationale in the response letter. The new version is even more interesting than the previous one, and much more grounded in the available data and analyses undertaken. All my comments below refer to the revised manuscript without track-changes.

Specific comments about the emphasis on co-evolution

In the response letter, the authors stated that “the objective of the analysis was only to demonstrate that the reviewed flatfish do not have a close phylogenetic relationship.” In lines 199-200, the authors wrote "… to test the hypothesis that a phylogenetic component of their hosts may influence ectoparasite community structure". I would replace it by something like "… to test how closely related the flatfish species are, so that hosts phylogeny can be considered as a factor explaining ectoparasite community structure". The current sentence “to test the hypothesis that a phylogenetic component of their hosts may influence ectoparasite community structure”, gives the impression that the authors are testing for co-phylogeny, but the authors themselves said in the response letter “… that the objective of our analyses was to investigate whether or not the assembly of parasite communities has a relationship with the phylogeny of the hosts, rather than the coevolution between parasites and specific hosts”, and that the fish phylogenetic trees lack resolution for them to make phylogenetic inferences. So, in my opinion the only modification needed is to downplay a little the “coevolution hypothesis” to match exactly what was explained in the response letter. For example, in L211-214, it reads “Subsequently, to check if the ectoparasite community structure of each host species had any congruence with the phylogenetic relationships of the fish, two analyses were carried out to group the fish based on the parasites species shared: one of similarity and another of parsimony.” I am not sure if “phylogenetic congruence” is the best wording here, maybe “to check if ectoparasite communities are more similar in closely related hosts than in hosts more phylogenetically distant”?

I also think the authors highlighted a valuable point in the response letter that is not mentioned in the text, and I quote “However, for us it was important that all

fish species reviewed in this study were represented in the phylogeny, and the only gene that

has been sequenced for all of them is cox 1.” I think a more polished version of this sentence could be added to the text, maybe in the methods or discussion. In addition to supporting the analyses in the way the authors have undertaken them (only with cox1), this sentence highlights a major gap in fish genetics.

Specific comments about grammar/sentence structure

L27 Remove comma before ‘presumably’ in “… in those species that, presumably”

L52 “Neady localities” – do you mean nearby?

L51-53 Please re-write this sentence to make it less repetitive “Thus, we would expect that in nearby localities, the similarity of the parasite communities in those localities would be high in neardy localities, but with increase in the geographical distance, it would decrease.”

One suggestion is “Thus, we would expect that in nearby localities, the similarity of the parasite communities would be high, decreasing with increased geographical distance”

L53 “Increase in the geographical distance” – replace by ‘increased geographical distance’

L.53 Remove “What is”

L182-183 “Additionally, an analysis of similarities (ANOSIM) was performed for the sampling dates.” – Add the reason why this was done to make it clearer for the reader. I suggest continuing the sentence with “to check for potential effects on parasite communities of sampling across different years.”

L203 Remove full stop after ‘Gulf of Mexico’

L246-248 Check grammar: ‘but not be representative’ should be ‘but are not representative’

L257-258 Check grammar and clarity in the following sentence: “However this effect seemed constant among regions and hosts since there were not significant intereactions were found between these factors”.

- Interactions instead of interactions

- Correct “there were not significant interactions were found”

- It is not clear what “this effect” and “these factors” refer to, please write in full instead of using “this” and “these”

The sentence the authors wrote in the response letter is actually much clearer, and could be used instead: “We found a significant potential effect of depth in parasite composition. However, this effect seemed to be constant among regions and hosts since there were not significant interactions with the other factors.”

L260 “… but no significant interaction was detected between regions and hosts”. This statement seems repetitive with the one in lines above (L257-258), but I can’t be sure because the statement in L 257-258 is confusing. Consider removing the repetition from L260 if redundant.

L335 (legend of Fig3): Remove bold font from the P in ‘Parsimony’

L360 “…parasite species give…” do you mean “are responsible for” or “are driving the results of similarity” instead of “give”?

L363 Instead of “species parasites” use “parasite species”

L428, 438, and 441 – Remove quotation marks from Conclusion

L712 Please translate the table legend to English

7. PLOS authors have the option to publish the peer review history of their article (what does this mean?). If published, this will include your full peer review and any attached files.

Reviewer #1: No

Reviewer #2: No

---

## [Author Response · Author response to Decision Letter 1]

28 Jun 2024

We thank the editor and reviewers for their recommendations and comments. All questions and comments made by the reviewers and the editor were considered in the "Responses to Reviewers R1" document.

---

## [Decision Letter · Decision Letter 2]

20 Aug 2024

Ecological and phylogenetic components of flatfish ectoparasites (Pleuronectiformes: Paralichthyidae) from the Southern Gulf of Mexico

PONE-D-23-31417R2

Dear Dr. Soler-Jimenez,

We’re pleased to inform you that your manuscript has been judged scientifically suitable for publication and will be formally accepted for publication once it meets all outstanding technical requirements.

Kind regards,

Joshua Kamani, PhD

Academic Editor

PLOS ONE

Additional Editor Comments (optional):

Dear Soler-Jimenez

I am happy to convey to you that your manuscript will be published subject to your making the corrections made by the reviewers. Kindly make the corrections and submit the final version on or before 30th August 2024. Thank you

Reviewers' comments:

Reviewer's Responses to Questions

**Comments to the Author**

1. If the authors have adequately addressed your comments raised in a previous round of review and you feel that this manuscript is now acceptable for publication, you may indicate that here to bypass the “Comments to the Author” section, enter your conflict of interest statement in the “Confidential to Editor” section, and submit your "Accept" recommendation.

Reviewer #1: All comments have been addressed

Reviewer #2: (No Response)

2. Is the manuscript technically sound, and do the data support the conclusions?

Reviewer #1: Yes

Reviewer #2: Yes

3. Has the statistical analysis been performed appropriately and rigorously? 

Reviewer #1: Yes

Reviewer #2: Yes

4. Have the authors made all data underlying the findings in their manuscript fully available?

Reviewer #1: Yes

Reviewer #2: Yes

5. Is the manuscript presented in an intelligible fashion and written in standard English?

Reviewer #1: Yes

Reviewer #2: No

6. Review Comments to the Author

Reviewer #1: The authors have appropriately addressed the comments and this manuscript is now acceptable for publication.

Reviewer #2: I commend the authors of the manuscript entitled “Ecological and phylogenetic components of flatfish ectoparasites” for their great work in improving clarity and coherence in this new version. My only suggestions are now regarding typographical errors, which are less frequent than in previous versions, but are still present. I list the ones I found below (referring to the revised manuscript without track-changes), but I would strongly recommend that more careful proof-reading is done prior to publication.

L43. “evolutive history” should be “evolutionary history”.

L84-85. “from north-western end” should be “from the north-western end”

L169. “the Bray-Curtis” should be “a Bray-Curtis”

L195-198 Unless two ANOSIM analyses were performed, this is repeated with L183-184. I suggest L195-198 could be removed, and L183-184 adapted to provide any extra information needed. For example, L183-184 could be “Additionally, an analysis of similarities (ANOSIM) was performed for the sampling dates to check for potential effects on parasite communities of sampling across different years, and no significant year effect was found (Table S3).”

L204. Replace 'and of which' with 'all of which'

L215. There are two ‘host’ that should be plural. Replace ‘closely related host’ by ‘closely related hosts’, and ‘distant host’ by ‘distant hosts’

L261. “any the other factors” should be “any of the other factors”

L313-314. Should “was the sister species of species of the genus” be “was the sister species of the genus” or “was the sister species of a species of the genus”?

L362. Remove “the” from “that the nonspecific parasite species”

L432. “GoMS” should be “SGoM”.

7. PLOS authors have the option to publish the peer review history of their article (what does this mean?). If published, this will include your full peer review and any attached files.

Reviewer #1: No

Reviewer #2: No

---

## [Editor Report · Acceptance letter]

26 Aug 2024

PONE-D-23-31417R2 

PLOS ONE

Dear Dr. Soler-Jiménez, 

I'm pleased to inform you that your manuscript has been deemed suitable for publication in PLOS ONE. Congratulations! Your manuscript is now being handed over to our production team.

Kind regards, 

on behalf of

Dr. Joshua Kamani 

Academic Editor

PLOS ONE